# Cross Cultivation on Homologous/Heterologous Plant-Based Culture Media Empowers Host-Specific and Real Time In Vitro Signature of Plant Microbiota

Hend Elsawey [1,2,†], Eman H. Nour [2,†], Tarek R. Elsayed [1,†], Rahma A. Nemr [1,†,‡], Hanan H. Youssef [1], Mervat A. Hamza [1], Mohamed Abbas [3], Mahmoud El-Tahan [4], Mohamed Fayez [1], Silke Ruppel [5] and Nabil A. Hegazi [1,*]

1   Department of Microbiology, Faculty of Agriculture, Cairo University, Giza 12613, Egypt
2   Faculty of Organic Agriculture, Heliopolis University for Sustainable Development, Cairo 11785, Egypt
3   Department of Microbiology, Faculty of Agriculture & Natural Resources, Aswan University, Aswan 81528, Egypt
4   Regional Center for Food and Feed, Agricultural Research Center, Giza 12613, Egypt
5   Department of Plant Microbe Systems, Leibniz Institute of Vegetable and Ornamental Crops, 14979 Grossbeeren, Germany
*   Correspondence: hegazinabil8@gmail.com; Tel./Fax: +20-2-35728483
†   These authors contributed equally to this work.
‡   Current address: Research and Development Department, Baramoda Company, Giza 12611, Egypt.

**Abstract:** Alliances of microbiota with plants are masked by the inability of in vitro cultivation of their bulk. Pure cultures piled in international centers originated from dissimilar environments/hosts. Reporting that plant root/leaf-based culture media support the organ-specific growth of microbiota, it was of interest to further investigate if a plant-based medium prepared from homologous (maize) supports specific/adapted microbiota compared to another prepared from heterologous plants (sunflower). The culture-independent community of maize phyllosphere was compared to communities cross-cultivated on plant broth-based media: CFU counts and taxa prevalence (PCR-DGGE; Illumina MiSeq amplicon sequencing). Similar to total maize phyllospheric microbiota, culture-dependent communities were overwhelmed by Proteobacteria (>94.3–98.3%); followed by Firmicutes (>1.3–3.7%), Bacteroidetes (>0.01–1.58%) and Actinobacteria (>0.06–0.34%). Differential in vitro growth on homologous versus heterologous plant-media enriched/restricted various taxa. In contrast, homologous cultivation over represented members of Proteobacteria (ca. > 98.0%), mainly Pseudomonadaceae and Moraxellaceae; heterologous cultivation and R2A enriched Firmicutes (ca. > 3.0%). The present strategy simulates/fingerprints the chemical composition of host plants to expand the culturomics of plant microbiota, advance real-time in vitro cultivation and lab-keeping of compatible plant microbiota, and identify preferential pairing of plant-microbe partners toward future synthetic community (SynComs) research and use in agriculture.

**Keywords:** plant-based culture media; culture-dependent/independent maize microbiota; in vitro host plant-specific cultivation; plant microbiota cross-host cultivation; homologous/heterologous cross cultivation



## 1. Introduction

Since the legendary work of L. Pasteur and R. Koch, and the introduction of "The Germ-Disease theory," culture media formulations and in vitro cultivation have marked the era of single colony isolation of pure isolates responsible for infectious diseases. Over the years, developments in culture media formulas and growth conditions/atmospheres resulted in breakthroughs in identifying key pure isolates of host-microbe holobionts [1–4]. Then, metagenomics founded the advanced era of exploring the broad spectra of diversity

of microorganisms beyond the culturable entities in their environments, and further portioning of core and satellite taxa from within [5–8]. Even with the breakthrough realized through OMIC approaches, a big gap still exists in understanding in vivo microbial gene functioning *in planta*, since many differentially expressed genes or gene families are not yet annotated [3]. Therefore, cultivation of microbial species is imperative and represents a major challenge to unearth the treasure of environmental microbiomes. In fact, pure cultures are key for studying microbial morphology, physiology, genomes, metabolomes, ecological impacts, and future manipulation/modification in the environment [3,4,9–13].

Along with, and to outpace the existing gap of unculturable communities, culturomics was introduced by Didier Raoult and Jean-Christophe Lagier group [14–16]. They were able to diversify the nutritional contents of various culture media together with the culturing conditions, atmospheres, and incubation times of human microbiota. This strategy enormously extended the known human gut microbiome, including archaea, to levels equivalent to those of the pyrosequencing repertoire [15,17]. In tandem, recent studies have combined genome sequencing with phenotypic characterization [18–20] and applied network-directed targeted bacterial isolation, reverse genomics, and genome-informed antibody for the cultivation of target specific groups of as-yet uncultured microbes [21].

Now, it is realized that conformity of in vitro with in vivo conditions is of prime importance to culture the unculturable members of environmental microbiota [4,12,13,22]. This was later implemented by the development of a number of strategies for in situ cultivation of microbes in their natural environments [10,23–28]. This made headway with the development of high-throughput methods of isolating several novel members of the untapped environmental microbiota. In addition, straightforward protocols were recently reported [3] for high-throughput bacterial isolation from plant roots using limiting dilution to ensure that most cultured bacteria originated from only one microorganism, followed by strain characterization and identification with the aid of an easy-to-use bioinformatics pipeline 'Culturome' (https://github.com/YongxinLiu/Culturome (accessed on 28 July 2022)) and a graphical user interface web server (http://bailab.genetics.ac.cn/culturome/ (accessed on 28 July 2022)). At the moment, there is a growing consensus among microbiologists that culture media are not a set list for laboratories to perform, but rather to be tailored in compatible with any of the tested environment/holobiont. This is in an effort to explore and unearth authentic oligotrophic taxa at the expense of high growth rate copiotrophics, i.e., to increase the cultivability of environmental microbiota and recover rare taxa [29–31].

Within this context, a number of strategies were introduced for in vitro cultivation of plant microbiota in culture media based on the sole use of plant materials in their original forms, which was reviewed by Sarhan et al. [4]. This is principally based on the use of crude plant saps, juices, and slurry homogenates [32,33], plant-broth [12,34,35], plant dehydrated powder teabags [36], and plant-based pellets [37]. Further, the direct use of intact leaves was introduced as culture pads [13] that provide a suitable environment and nutrients in their natural composition and complexity for co-culturing and to gain insight into interspecific/intraspecific interactions among culturable communities [13,38,39]. Such plant materials provide diverse plant macromolecules, major and minor elements, and growth factors in the form of amino acids and other compounds of unknown composition and concentration. Accordingly, such culturing strategies create an "in-situ-similis" vegan nutritional matrix that favors in vitro cultivability of plant microbiota very much similar to conditions *in planta*. Pooling the advantages of MPN enrichment, this strategy is further extended by exploring plant organ-compatible cultivation of the microbiota of sunflower when cross-cultivated on corresponding leaf/root-based culture media. PCR-DGGE analyses and pure isolate 16S rRNA sequencing indicated divergence in the community composition of cultivable endophytes of plant compartments, e.g., signaling a certain degree of plant organ affinity/compatibility [39]. Those patterns of microbial community assembly were reported to be directly related to the chemical composition and succession in the plant, within its various compartments, and in its vicinity as root exudates [40], which vary in quantity and quality with plant species, genotype, age, and physiological status.

Overall, this creates an in vitro nutritional assemblage compatible with the nutritional needs of host-innate microbiota [41,42].

The advancement of plant-only-based culturing strategies paved the way to expand the culturability of plant microbiota, to recover slow-growing microorganisms [43], and to obtain isolates of not-yet cultured genera and less abundant and/or hard-to-culture bacterial phyla representing novel and rapidly increasing candidate phyla [4,12,13,39]. Furthermore, such culturing strategies have been successfully adjusted for microbial biomass production on pilot/industrial levels [34,37,44]. No doubt that progress of in vitro isolation and domestication of plant microbiota opens future prospects of application in the field, where possibilities exist of future introductions to modify the microbiota embracing plants in the form of in vitro-synthetic core formulas (SynComs). A potential approach to support future agriculture under stressed environments exposed to serious climatic changes.

Our previous report on the divergence in community composition of cultivable endophytes of sunflower that signaled a certain degree of plant organ (leaves/roots) affinity/compatibility [39] encouraged the present efforts to investigate diversity exposure by in vitro cross cultivation among host plants. Taking into consideration that the chemical diversity of plants is very high, as plants produce a diverse array of lineage-specific specialized (secondary) metabolites that are synthesized from primary metabolites [45,46]. Reports have indicated that such a huge array of metabolites are more than those produced by most other organisms and have provided deeper insights into the genetic bases of such metabolic diversity at both population and individual levels [45,47,48]. Such diverse metabolites play many different roles in plant growth and development in response to continually changing environmental conditions, as well as abiotic/biotic stresses. To a large extent, this is attributed to the chemical modification of the basic skeletons of metabolites [49]. Here, plant- and microbial-specialized metabolites are intermingled, constitute immense chemical diversity, and play key roles in mediating ecological interactions between organisms [50]. Therefore, simulating such complex nutritional matrices into in vitro tailored culture media is a real challenge that justifies abandoning the use of chemically synthetic culture media. It also compels the use of plant culture media based on compatible/homologous host plants with their specific chemical composition and complexity fingerprints.

In the present study, the culturable community composition of plant microbiota was studied using culture media based on the tested homologous host plant (maize plants) in comparison to another heterologous host plant (sunflower plants). This was expressed in terms of: (a) population densities in the form of colony-forming units (CFUs), (b) diversity structure of the culture-dependent communities measured by PCR-DGGE fingerprinting of the 16S rRNA gene segment, and (c) Illumina MiSeq sequencing and analysis of 16S rRNA gene amplicons from total community DNA (TC-DNA). This is in comparison with the culture-independent analyses of representative samples of compartments of maize plants, endo-rhizosphereand endo-phyllosphere.

## 2. Materials and Methods

### 2.1. Hypothesis and Experimental Design

The main objective of the study was to report on the appropriateness of in vitro cross cultivation of plant microbiota on plant-only culture media based on different host plants (Graphical Abstract). For this purpose, the cultivability of maize endophytes was tested on culture media based on plant broth prepared from the selfsame host plant (maize, i.e., homologous combination) compared to the plant broth of a different host plant (sunflower, i.e., heterologous combination), as well as the chemically-synthetic standard culture medium of R2A. The culturable community developed on agar plates was followed in the form of CFUs, and community composition was assessed using DGGE and Illumina MiSeq amplicon sequencing analyses. For comparison, metagenomic samples of the tested plant compartments (endo-rhizosphere and endo-phyllosphere) were included in the analysis.

### 2.2. Tested Plant Materials

The tested host plants of maize (*Zea mays* L.) and sunflower (*Helianthus annuus* L.) were grown in the open fields of the experimental station, Faculty of Agriculture, Cairo University, Giza, Egypt (30.0131° N, 31.2089° E). Representative samples of shoots (leaves and young stems) and roots of either full-grown plant were collected in plastic bags [13]. The samples were brought to the laboratory and kept in a refrigerator prior to further processing.

### 2.3. Plant Broth

According to Elsawey et al. [12], coarse-chopped plant shoots of maize and sunflower were washed and soaked in 10 L-Erlenmeyer flasks with tap water (1:2, *w/v*). After heat extraction in an autoclave (121 °C for 20 min), the mixture was pressed and cross-filtered through cotton cloth to collect the cleared broth. Broths were distributed in aliquots and stored at −20 °C till use.

### 2.4. Culture Media
#### 2.4.1. Plant Broth-Based Culture Media (Elsawey et al. [12])

The plant broth culture media was prepared by the addition of different volumes (*v/v*) of prepared plant broth to distilled water (5 and 25 mL L$^{-1}$). Agar culture media were prepared by the addition of agar (2% *w/v*), pH was kept as such without adjustment, and it was in the range of 6.0–6.8; then autoclaved at 121 °C for 20 min.

#### 2.4.2. Chemically-Synthetic Standard Culture Medium

R2A agar medium was used with a slight modification that contained (g L$^{-1}$): casein hydrolysate, 0.5; dextrose, 0.5; soluble starch, 0.5; yeast extract, 0.5; dipotassium phosphate, 0.3; sodium pyruvate, 0.3; casein peptone, 0.25; meat peptone, 0.25; and magnesium sulfate, 0.024. Agar was added (2% *w/v*), and the pH was adjusted to 7.0 ± 0.2 [12,51]; https://assets.fishersci.com/TFS-Assets/LSG/manuals/IFU112543.pdf (accessed on 29 September 2020).

### 2.5. In Situ Recovery and Cultivability of Endophytes of Maize Endo-Rhizosphere and Endo-Phyllosphere

For preparation of the endo-rhizosphere and endo-phyllosphere, root/leaf samples of maize were initially washed and surface sterilized according to Youssef et al. [52] for roots and de Oliveira Costa et al. [53]; Jackson et al. [54] for leaves. From this original root/leaf suspensions (5 g root/leaf in 45 mL basal salts of CCM culture medium [55] as a diluent, referred to as the mother culture), further serial dilutions were prepared. Aliquots (200 µL) of suitable dilutions were surface inoculated on agar plates, with four replicates prepared from all of the tested culture media. Incubation took place at 25 °C for 1–14 days, and CFUs, including micro-colonies (µCo, <1 mm diameter discriminated with 40× magnification), were observed and counted throughout [12]. Dry weights of roots/leaves were obtained by drying the original roots/leaves suspensions at 70 °C for 1–2 days.

### 2.6. In Situ Diversity of Culturable Endophytes of Maize Developed on Homologous (Maize) and Heterologous (Sunflower) Plant Broth

In situ recovery and cultivation of endophytic bacteria of maize compartments, rhizosphere, and phyllosphere were performed on homologous (maize) and heterologous (sunflower) plant broth-based culture media using two concentrations of plant broth (5- and 25-mL L$^{-1}$) (Table S1). For culturable community composition, all CFUs developed on representative agar plates, after long incubation for 14 days, were harvested for DNA extraction, DGGE, and Illumina MiSeq sequencing analyses.

### 2.7. DNA Extraction

For DNA extraction, and according to Sarhan et al. [36], all CFUs developed on representative long-incubated agar plates (>30–300 CFUs plate$^{-1}$) of the tested culture

media were harvested using 0.05 M NaCl solution. DNA was also extracted from the initial root and leaf suspensions originally prepared for CFU counting (referred to as the mother culture). Aliquots of 2 mL of both harvested CFUs and suspension of mother cultures were centrifuged for 10 min at 10,000 rpm. DNA was extracted from collected pellets using the genomic DNA Extraction Mini Kit (iNtRON Biotechnology, Kyungki-Do, Korea) according to the manufacturer's instructions. DNA quality was assessed using a NanoPhotometer (NanoPhotometer NP80 Touch, Implen GmbH, Munich, Germany).

## 2.8. Amplification of the 16S rRNA Gene and DGGE Fingerprinting

Total community DNA (TC-DNA) extracted from CFU harvest, as well as mother cultures, was used to amplify the whole 16S rRNA gene using the 9bfm (GAGTTTGATY-HTGGCTCAG) and 1512r (ACGGHTACCTTGTTACGACTT) primers [39,56]. The reaction was performed using a Hightech Thermocycler Cycler (SensoQuest, Göttingen, Germany) in a total volume of 25 μL with 2 μL template DNA (ca. 2–18 ng μL$^{-1}$), 12.5 μL of QIAGEN TopTaq master mix (Qiagen Inc., Hilden, Germany), 5.5 μL PCR water, and 2.5 μL of 3.3 pmol of both primers. The thermal cycling program was adjusted as follows: 4 min initial denaturation at 95 °C, 30 thermal cycles of 1 min denaturation at 95 °C, 1 min annealing at 56 °C, and 1 min extension at 74 °C; PCR was completed by a final extension step at 74 °C for 10 min. A QIAquick PCR purification kit (Qiagen Inc., Hilden, Germany) was used to purify the PCR product according to the manufacturers' instructions. To obtain the PCR product of the V3 region, 2 μL of the purified 16S rRNA PCR product (10 ng μL$^{-1}$) were re-amplified using the 341f-GC (CGCCCGCCGCGCGCGGCGGGGCGGGGGCGGGGGCACGGGGCCTACGGGAGGC-AGCAG) and 518r (ATTACCGCGGCTGCTGG) primers [56,57]; the reaction conditions and thermal cycling program were used as described above.

PCR products of the V3 region were heated at 95 °C for 5 min and stored at 65 °C before loading onto the gradient gel. The products of both PCR reactions were tested on a 1.5% agarose gel to ensure a single product of the expected size.

DGGE was performed using the VS20WAVE-DGGE Mutation Detection System (Cleaver Scientific, United Kingdom) [36,39]. PCR products (10 μL of 10–15 ng PCR products mixed with 3 μL 6X loading dye) were electrophoresed on 8% polyacrylamide gel containing 30% to 70% denaturing gradient of formamide and urea with 1x TAE buffer. DGGE was conducted at 60 °C for 20 hrs at a voltage of 50 V. The gel was stained for 30 min with a 6X Ethidium bromide stain and recorded with a UV Transilluminator (Cleaver Scientific, United Kingdom). A self-created standard of mixed PCR products from four pure bacterial strains (*Listeria innocua* DSM 20649, *Arthrobacter globiformis* DSM 20124, *Lactobacillus plantarum* DSM 20174, and *Bifidobacterium breve* DSM 20213) was included in every DGGE run. All of these strains were obtained from DSMZ-Germany (dsmz.de) and revived according to the provider's instructions.

The DGGE fingerprints were analyzed using CLIQS (TotalLab, Newcastle upon Tyne, UK). The total number of DGGE bands was used to represent the 16S rRNA gene assortment.

## 2.9. Illumina MiSeq Sequencing and Analysis of 16S rRNA Gene Amplicons from TC-DNA

A total of 17 DNA samples (Table S1) were subjected to a paired-end read Illumina MiSeq platform targeting the V3–V4 region of the 16S rRNA gene, using the 515f/806r primer set by ATLAS Biolabs GmbH, Berlin, Germany.

The sequencing data were uploaded to the Galaxy web platform using the public server at usegalaxy.org (accessed on 1 April 2022) to analyze the data [58] with default settings (based on the Standard Operating Procedure (SOP)) for MiSeq data [59]. Paired-end mating was applied with a minimum overlap length of 50 bp, maximum mismatches of 15, and a minimum quality of 30. Operational taxonomic units (OTUs) were picked at a 97% sequence identity level. The OTUs' representative sequences were selected by the highest abundance within the cluster and assigned to taxonomy using the GreenGene classifier [60]. Sequence counts of each sample are provided (Table S1), and a rarefaction curve is presented

(Figure S1). Community analysis was performed using the statistical analysis of taxonomic and functional profiles (STAMP) software 2.1.3 [61]. Significant changes in the relative abundance of dominant taxa were identified with ANOVA, followed by Tukey's honest significance detection test ($p < 0.05$) [62]. Sequences were submitted for deposition at the public repository Sequence Read Archive (SRA) with accession number PRJNA900715 and bioproject accession number PRJNA891051 (http://www.ncbi.nlm.nih.gov/sra (accessed on 12 November 2022)).

The linear discriminant analysis (LDA) effect size (LEfSe) method was performed using the online platform (Galaxy (harvard.edu), http://huttenhower.sph.harvard.edu/lefse/ (accessed on 14 August 2022) according to Segata et al. [63]. Operational taxonomic units (OTUs) obtained from amplicon sequencing data of both culture-dependent and culture-independent samples were used to explain differences between classes by coupling standard tests for statistical significance with additional tests encoding biological consistency and effect relevance. The aim of this analysis is to predict groups of organisms or operational taxonomic units that concisely differentiate the classes being compared, i.e., to highlight and discover metagenomic biomarkers.

### 2.10. Chemical Analysis of Dehydrated Plant Powders

The chemical compositions and nutritional contents of the tested plants (maize and sunflower) were determined by the certified Regional Center for Food and Feed (RCFF), Agricultural Research Center (ARC), Giza, Egypt ([12]; https://psm.gov.eg/providers/213/services (accessed on 16 July 2022)). Analyses included total crude protein, total crude fiber, total ash, total carbohydrates, amino acids, macronutrients, and micronutrients (Table S2).

### 2.11. Statistical Analysis

For CFU counts, the analysis of variance and the least significant differences (LSD) were calculated using MSTAT-C software Michigan State University, East Lansing, MI, USA) to examine the independent effects of incubation period, plant sphere, and culture media besides their interactions. Used as well are the R-project packages (cran.r-project.org (accessed on 10 May 2022): "agricolae" for statistical analysis and "ggplot2" for constructing boxplots.

### 3. Results

### 3.1. In Situ Diversity of Culturable Endophytes of Maize Developed on Culture Media Based on Homologous Broth of Maize and Heterologous Broth of Sunflower

Colony-forming units (CFUs) of culturable endophytes associated with maize rhizo- and phyllo-spheres were nicely developed on all tested culture media (Figure S2). Regarding CFU counts, ANOVA analysis indicated significant differences attributed to the single effects of the incubation period and plant compartments (Table 1). The total numbers of CFUs were significantly increased (>5% increases) with the increase of incubation time, being in the range of ca. log $> 6.0$–$7.0$ CFUs g$^{-1}$DW. The endophytic load of the plant endo-phyllosphere was significantly lower (>15%), and approximated one log CFUs g$^{-1}$ DW compared to that of the endo-rhizosphere (Figure S3). Irrespective of the plant sphere, ANOVA one-way (Table 1) and 2-way interactions (Table S3) of culture media and incubation time indicated that the nutritional store of all tested culture media are affluent to support indiscriminate development of CFUs. Counts were within the range of log 6.5 to log 7.6 CFUs g$^{-1}$DW, and the differences approximated 10%. Counts of CFUs developed on all of the tested plant-based culture media (log 6.87–log 7.60 CFUs g$^{-1}$DW) were very much proportionate to those reported for the chemically-synthetic R2A standard culture medium (log 7.47 CFUs g$^{-1}$DW).

**Table 1.** One-way ANOVA analysis of log numbers of CFUs (data are log means $\pm$ standard error [SE], *n* = 5) of endophytes of the endo-rhizosphere and endo-phyllosphere of maize developed on all tested culture media.

| Treatments | Log No. CFUs g$^{-1}$DW. | |
|:---:|:---:|:---:|
| | **Total Colonies** | **Micro-Colonies** |
| **Incubation time** | | |
| **1 day** | 6.96 $\pm$ 0.090 [c] | 6.01 $\pm$ 0.300 [a] |
| **4 days** | 7.28 $\pm$ 0.104 [b] | 4.36 $\pm$ 0.304 [b] |
| **7 days** | 7.30 $\pm$ 0.105 [ab] | 3.67 $\pm$ 0.300 [b] |
| **14 day** | 7.33 $\pm$ 0.107 [a] | 3.63 $\pm$ 0.300 [b] |
| **LSD (*p* value $\leq$ 0.05)** | **0.041** | **0.84** |
| **Plant sphere** | | |
| **Endo-rhizosphere** | 7.77 $\pm$ 033 [a] | 6.20 $\pm$ 0.214 [a] |
| **Endo-phyllosphere** | 6.67 $\pm$ 0.059 [b] | 2.69 $\pm$ 0.212 [b] |
| **LSD (*p* value $\leq$ 0.05)** | **0.029** | **0.60** |
| **Culture medium** | | |
| **R2A** | 7.47 $\pm$ 0.020 [b] | 2.85 $\pm$ 0.336 [b] |
| **MPhYsH** | 7.60 $\pm$ 0.089 [a] | 3.13 $\pm$ 0.341 [b] |
| **MPhYsL** | 7.36 $\pm$ 0.089 [c] | 5.18 $\pm$ 0.336 [a] |
| **MPhYmH** | 6.79 $\pm$ 0.127 [e] | 5.10 $\pm$ 0.336 [a] |
| **MPhYmL** | 6.87 $\pm$ 0.148 [d] | 5.82 $\pm$ 0.336 [a] |
| **LSD (*p* value $\leq$ 0.05)** | **0.06** | **0.94** |

R2A, Reasoner's 2A agar; MPhYmH, culture-dependent on homologous maize broth 25 ml L$^{-1}$; MPhYmL, culture-dependent on homologous maize broth 5 ml L$^{-1}$; MPhYsH culture-dependent on heterologous sunflower broth 25 ml L$^{-1}$; MPhYsL, culture-dependent on heterologous sunflower broth 5 ml L$^{-1}$. Statistically significant differences are designated by different letters ($p \leq 0.05$, *n* = 5).

Of interest was the domination of micro-colonies that in average represented >40–85% of the total colonies developed on agar plates. Their percentages significantly decreased with prolonged incubation time, being less than 40% at 14 days of incubation (Table 1). The percentages of such micro-colonies were significantly higher in the endo-rhizosphere (average of ca. 70%) compared to the endo-phyllosphere (average of ca. 45%). The distinctive and confined growth of colonies, being less slimy and/or watery, developed on agar plates of plant-only-based culture media permitted the particular development of such micro-colonies (>40–85%) in comparison to the chemically-synthetic R2A culture medium (40–58%) (Figure S2).

*3.2. Divergence in Culturable Community Composition of Maize Bacterial Endophytes as Indicated by DGGE Analysis*

PCR-DGGE fingerprinting of the 16S rRNA gene segment recovered from CFUs developed on tested culture media was performed to compare the composition of the cultivable endo-rhizosphere and endo-phyllosphere bacterial communities of maize. Included in the analysis were the genomic 16S rRNA samples that were extracted from the initial leaf and root suspensions, originally prepared for culture-dependent analysis (referred to as the mother culture). The UPGMA analysis resulted in clear banding patterns and clustering of the produced DGGE bands of endophytic bacteria in both plant compartments. For the endo-phyllosphere community and based on the analysis of distance scores, the UPGMA clustering was differentiated into two main clusters; one separated earlier at a cluster cutoff value of 0.57, for the mother culture-independent maize leaf and its first serial dilution (Figure 1). Later, at a cluster cutoff value of 0.86, the tested culture media were further

sub-clustered, where the culturable community developed on homologous maize plant broth distanced from those of heterologous sunflower plant broth together with the standard R2A culture medium. A somewhat similar assortment was reported for the bacterial communities of the maize endo-rhizosphere (Figure S4).

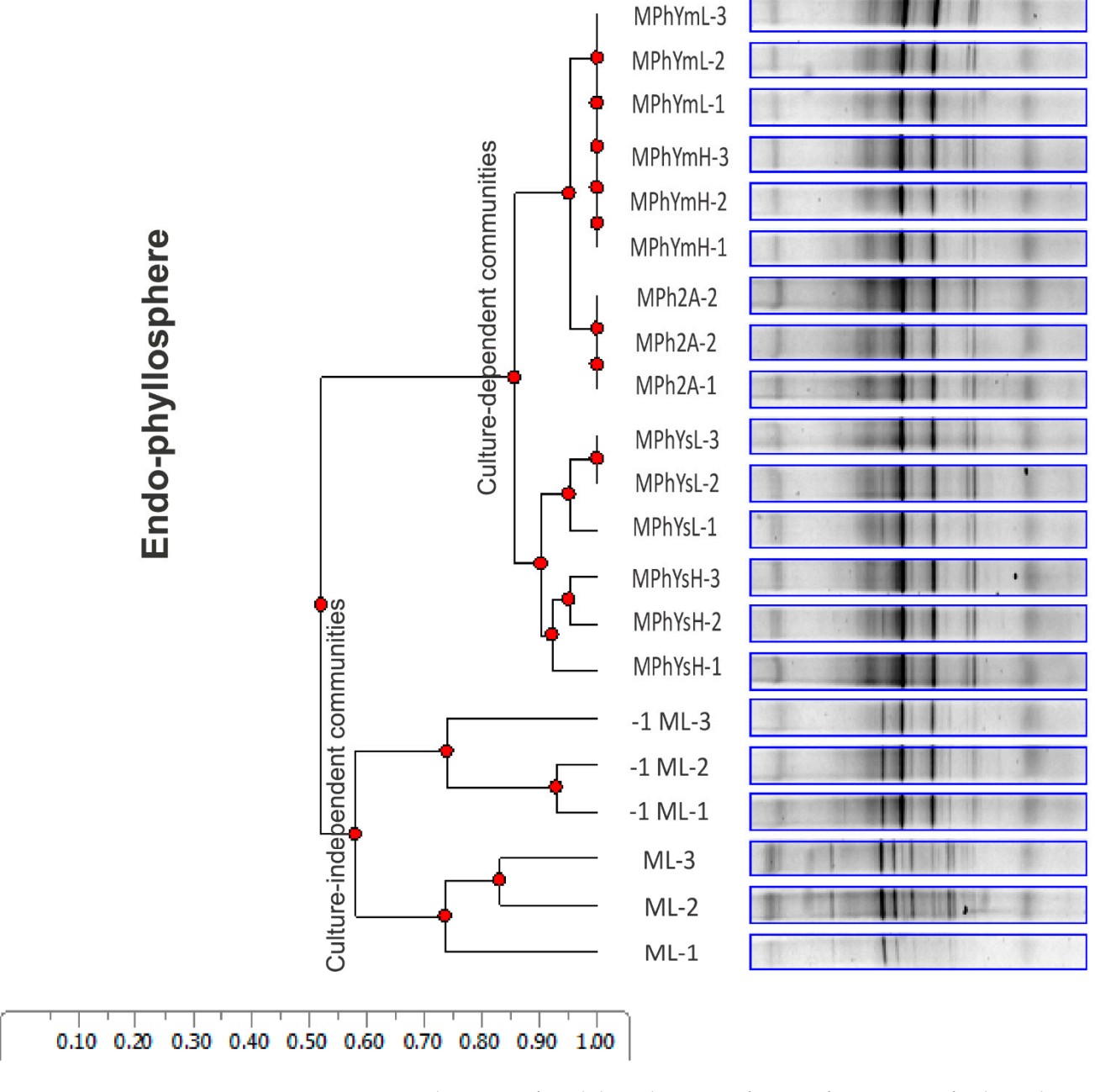

**Figure 1.** UPGMA clustering of Euclidean distances of DGGE fingerprints of culture-dependent and culture-independent maize endo-phyllosphere bacterial communities. Each culture medium and endo-phyllosphere mother culture is represented by three replicates (plates 1–3). ML, culture-independent mother leaf; MPhYmH, 1ML, 1/10 dilution of culture-independent mother leaf; MPhYmH, culture-dependent on homologous maize broth 25 mL L$^{-1}$; MPhYmL, culture-dependent on homologous maize broth 5 mL L$^{-1}$; MPhYsH culture-dependent on heterologous sunflower broth 25 mL L$^{-1}$; MPhYsL, culture-dependent on heterologous sunflower broth 5 mL L$^{-1}$; MPh2A, culture-dependent of R2A culture medium.

### 3.3. Amplicon Sequence Data Analysis of Maize Endo-Phyllosphere Bacteria

#### 3.3.1. Total OTUs Obtained

Illumina MiSeq paired-end amplicon sequencing of V3–V4 regions of the 16S rRNA gene was obtained from TC-DNA extracted from culture-dependent and -independent samples. After sequence quality screening, a total of 1,405,682 bacterial sequences were generated from the 17 tested samples. On average, OTUs recorded were in the range of >70,000–90,000 (Table S1). The obtained bacterial sequences were affiliated into 9 phyla, 16 classes, 30 orders, 51 families, and 67 genera. The taxonomic affiliation of representative OTUs is presented in Section 3.3.3 and Table 2.

**Table 2.** Relative abundance of maize endophytes as dominant phyla, classes, and orders in all tested culture-independent and culture-dependent samples (average ± standard error of the mean).

| Phylum | ML | MPh2A | MPHYmH | MphYmL | MphYsH | MphYsL |
|---|---|---|---|---|---|---|
| Proteobacteria | 94.81 ± 2.16 [ab] | 94.76 ± 1.32 [ab] | 98.26 ± 0.25 [a] | 98 ± 0.41 [ab] | 95.27 ± 2.26 [ab] | 94.32 ± 0.82 [b] |
| Firmicutes | 1.63 ± 0.38 [bc] | 3.73 ± 0.5 [a] | 1.25 ± 0.22 [c] | 1.43 ± 0.32 [c] | 3.25 ± 0.8 [ab] | 3.34 ± 0.6 [a] |
| Bacteroidetes | 3.09 ± 2.82 | 0.96 ± 1.66 | 0 ± 0 | 0.01 ± 0.01 | 0.92 ± 1.49 | 1.58 ± 0.4 |
| Actinobacteria | 0.11 ± 0.15 | 0.12 ± 0.09 | 0.06 ± 0.04 | 0.07 ± 0.03 | 0.17 ± 0.19 | 0.34 ± 0.29 |
| **Class** | **ML** | **MPh2A** | **MphYmH** | **MphYmL** | **MphYsH** | **MphYsL** |
| Gammaproteobacteria | 66.06 ± 12.71 [b] | 79.61 ± 1.55 [ab] | 85.8 ± 2.73 [a] | 84.78 ± 3.85 [a] | 77.02 ± 1.39 [ab] | 72.45 ± 3.06 [b] |
| Alphaproteobacteria | 21.34 ± 4.12 [a] | 14.75 ± 1.1 [ab] | 12.35 ± 2.82 [b] | 12.49 ± 4.33 [b] | 16.98 ± 0.83 [ab] | 18.43 ± 2.16 [ab] |
| Betaproteobacteria | 7.3 ± 6.42 [a] | 0.3 ± 0.51 [b] | 0 ± 0 [b] | 0.6 ± 0.54 [b] | 1.16 ± 0.92 [ab] | 3.33 ± 1.58 [ab] |
| Proteobacteria_unclassified | 0.11 ± 0.02 | 0.1 ± 0.01 | 0.1 ± 0.01 | 0.12 ± 0.03 | 0.1 ± 0 | 0.11 ± 0.02 |
| Bacilli | 1.47 ± 0.15 [b] | 3.73 ± 0.5 [a] | 1.25 ± 0.22 [b] | 1.43 ± 0.32 [b] | 3.25 ± 0.8 [a] | 3.33 ± 0.62 [a] |
| Clostridia | 0.16 ± 0.23 | 0 ± 0 | 0 ± 0 | 0 ± 0 | 0 ± 0 | 0 ± 0 |
| Flavobacteria | 3.09 ± 2.82 | 0.95 ± 1.65 | 0 ± 0 | 0 ± 0 | 0.87 ± 1.4 | 1.53 ± 0.43 |
| Saprospirae | 0 ± 0 | 0 ± 0 | 0 ± 0 | 0 ± 0 | 0.05 ± 0.08 | 0 ± 0 |
| Sphingobacteria | 0 ± 0 | 0 ± 0 | 0 ± 0 | 0 ± 0 | 0 ± 0 | 0.05 ± 0.09 |
| Actinobacteria | 0.11 ± 0.15 | 0.12 ± 0.09 | 0.06 ± 0.04 | 0.07 ± 0.03 | 0.17 ± 0.19 | 0.34 ± 0.29 |
| **Order** | **ML** | **MPh2A** | **MphYmH** | **MphYmL** | **MphYsH** | **MphYsL** |
| Pseudomonadales | 38.24 ± 4.92 [c] | 72.78 ± 2.2 [ab] | 77.24 ± 3.61 [a] | 78.37 ± 0.59 [a] | 71.98 ± 1.11 [ab] | 66.99 ± 3.51 [b] |
| Enterobacteriales | 22.68 ± 7.92 [a] | 4.6 ± 0.84 [b] | 5.84 ± 0.72 [b] | 3.37 ± 3.69 [b] | 2.81 ± 1.74 [b] | 2.94 ± 1.2 [b] |
| Gammaproteobacteria_unclassified | 1.05 ± 0.07 [b] | 2.21 ± 0.18 [ab] | 2.72 ± 0.19 [a] | 3.03 ± 0.62 [a] | 2.05 ± 0.25 [ab] | 2.47 ± 0.56 [a] |
| Xanthomonadales | 4.09 ± 0.2 [a] | 0.01 ± 0.01 [b] | 0.01 ± 0 [b] | 0.01 ± 0 [b] | 0.17 ± 0.27 [b] | 0.05 ± 0.02 [b] |
| Rhizobiales | 15.21 ± 4.58 [a] | 0.45 ± 0.35 [b] | 0.14 ± 0.16 [b] | 1.91 ± 3.11 [b] | 0.76 ± 0.51 [b] | 1.3 ± 0.73 [b] |
| Sphingomonadales | 6.11 ± 0.46 [bc] | 14.12 ± 1.2 [ab] | 11.96 ± 2.3 [ab] | 10.55 ± 3.78 [bc] | 16.17 ± 0.88 [a] | 17.09 ± 2.14 [a] |
| Alphaproteobacteria_unclassified | 0.02 ± 0 | 0.03 ± 0 | 0.03 ± 0.01 | 0.02 ± 0.01 | 0.04 ± 0 | 0.04 ± 0.01 |
| Caulobacterales | 0 ± 0 | 0.13 ± 0.23 | 0 ± 0 | 0 ± 0 | 0.01 ± 0.01 | 0 ± 0 |
| Rhodobacterales | 0 ± 0 | 0.01 ± 0.01 | 0.22 ± 0.37 | 0.01 ± 0 | 0.01 ± 0.01 | 0 ± 0 |
| Burkholderiales | 7.24 ± 6.35 [a] | 0.29 ± 0.5 [b] | 0 ± 0 [b] | 0.59 ± 0.53 [b] | 1.15 ± 0.91 [ab] | 3.3 ± 1.58 [ab] |
| Betaproteobacteria_unclassified | 0.06 ± 0.07 | 0 ± 0.01 | 0 ± 0 | 0.01 ± 0.01 | 0.01 ± 0.01 | 0.03 ± 0.01 |
| Proteobacteria_unclassified | 0.11 ± 0.02 | 0.1 ± 0.01 | 0.1 ± 0.01 | 0.12 ± 0.03 | 0.1 ± 0 | 0.11 ± 0.02 |
| Bacillales | 1.47 ± 0.15 | 3.73 ± 0.5 | 1.25 ± 0.22 | 1.43 ± 0.32 | 3.25 ± 0.8 | 3.33 ± 0.62 |
| Clostridiales | 0.16 ± 0.23 | 0 ± 0 | 0 ± 0 | 0 ± 0 | 0 ± 0 | 0 ± 0 |
| Actinomycetales | 0.11 ± 0.15 | 0.12 ± 0.09 | 0.06 ± 0.04 | 0.07 ± 0.03 | 0.17 ± 0.19 | 0.34 ± 0.29 |

Number shows the average percentage followed by ± standard deviation. Treatments sharing the same letters are non-significantly different (*p* < 0.05, ANOVA under generalized linear model followed by Tukey's Honest Significant Detection test). Significant increases in abundance compared to the ML are highlighted in green, while significant decreases are highlighted in red. ML, culture-independent mother leaf; MPhYmH, culture-dependent on homologous maize broth 25 mL L$^{-1}$; MPhYmL, culture-dependent on homologous maize broth 5 mL L$^{-1}$; MPhYsH culture-dependent on heterologous sunflower broth 25 mL L$^{-1}$; MPhYsL, culture-dependent on heterologous sunflower broth 5 mL L$^{-1}$; MPh2A, culture-dependent of R2A culture medium.

### 3.3.2. UPGMA and PCA Analyses

Cluster dendrogram analysis (UPGMA) based on the relative abundance of all bacterial OTUs revealed two major distinct clusters at the early separation level: one was confined to the culture-independent mother samples, while the second included all of the culture-dependent samples (Figure 2). Further, the culturable bacterial communities were distinctly divided into three sub-clusters, based on the used culture media: a, the heterologous sunflower plant broth; b, homologous maize plant broth; and c, standard chemically synthetic R2A. In addition, PCA analysis indicated the distinct separation among the culturable communities developed on the three different culturing methods in relation to the culture-independent community at the phylum, order, and genus levels (Figure 3).

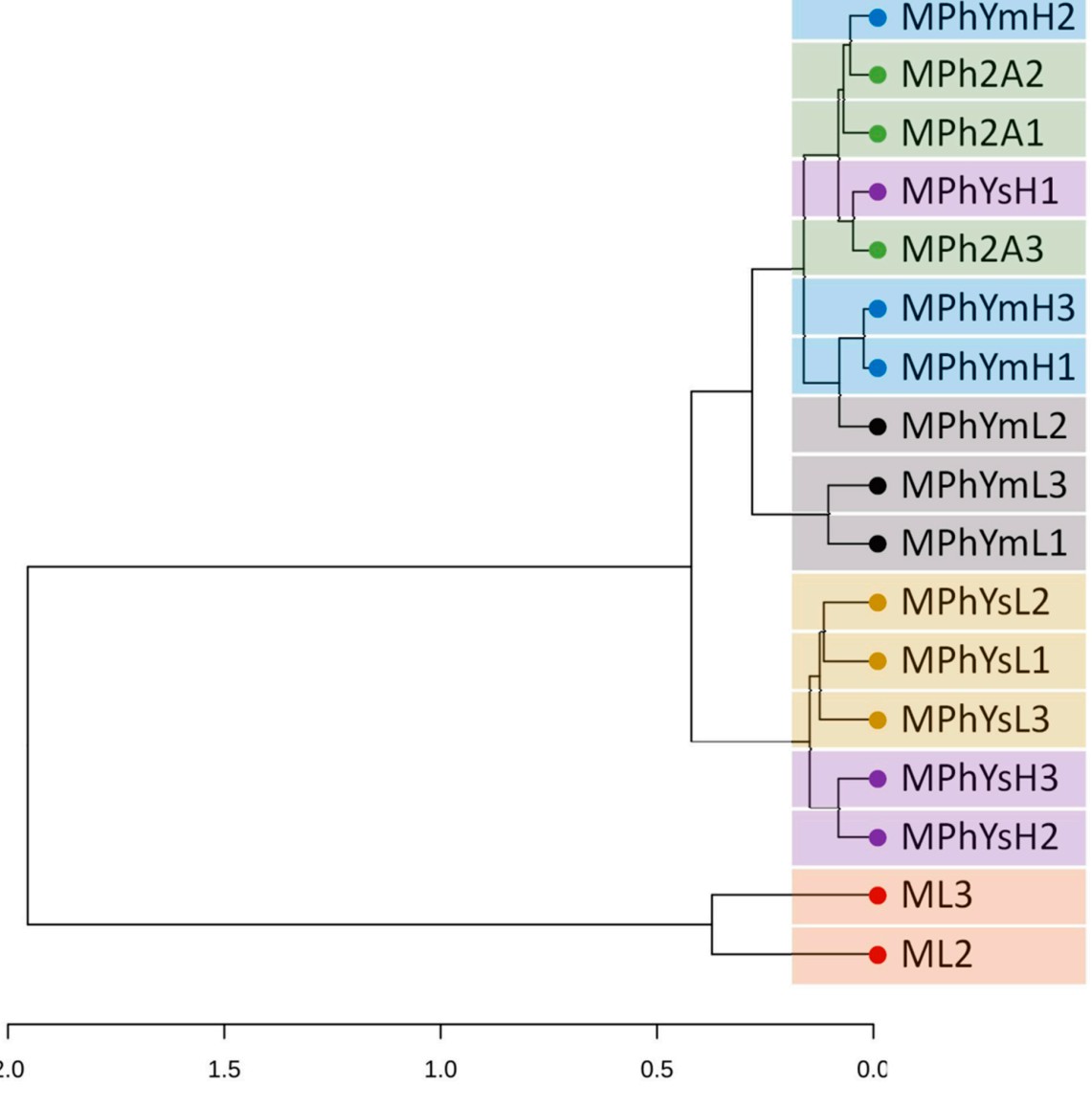

**Figure 2.** Based on MiSeq sequencing data, cluster analysis of the bacteria community based on all OTUs detected for culture-independent/dependent samples (clustering method = UPGMA, distance = Euclidean). ML, culture-independent mother leaf; MPhYmH, culture-dependent on homologous maize broth 25 mL L$^{-1}$; MPhYmL, culture-dependent on homologous maize broth 5 mL L$^{-1}$; MPhYsH culture-dependent on heterologous sunflower broth 25 mL L$^{-1}$; MPhYsL, culture-dependent on heterologous sunflower broth 5 mL L$^{-1}$; MPh2A, culture-dependent of R2A culture medium.

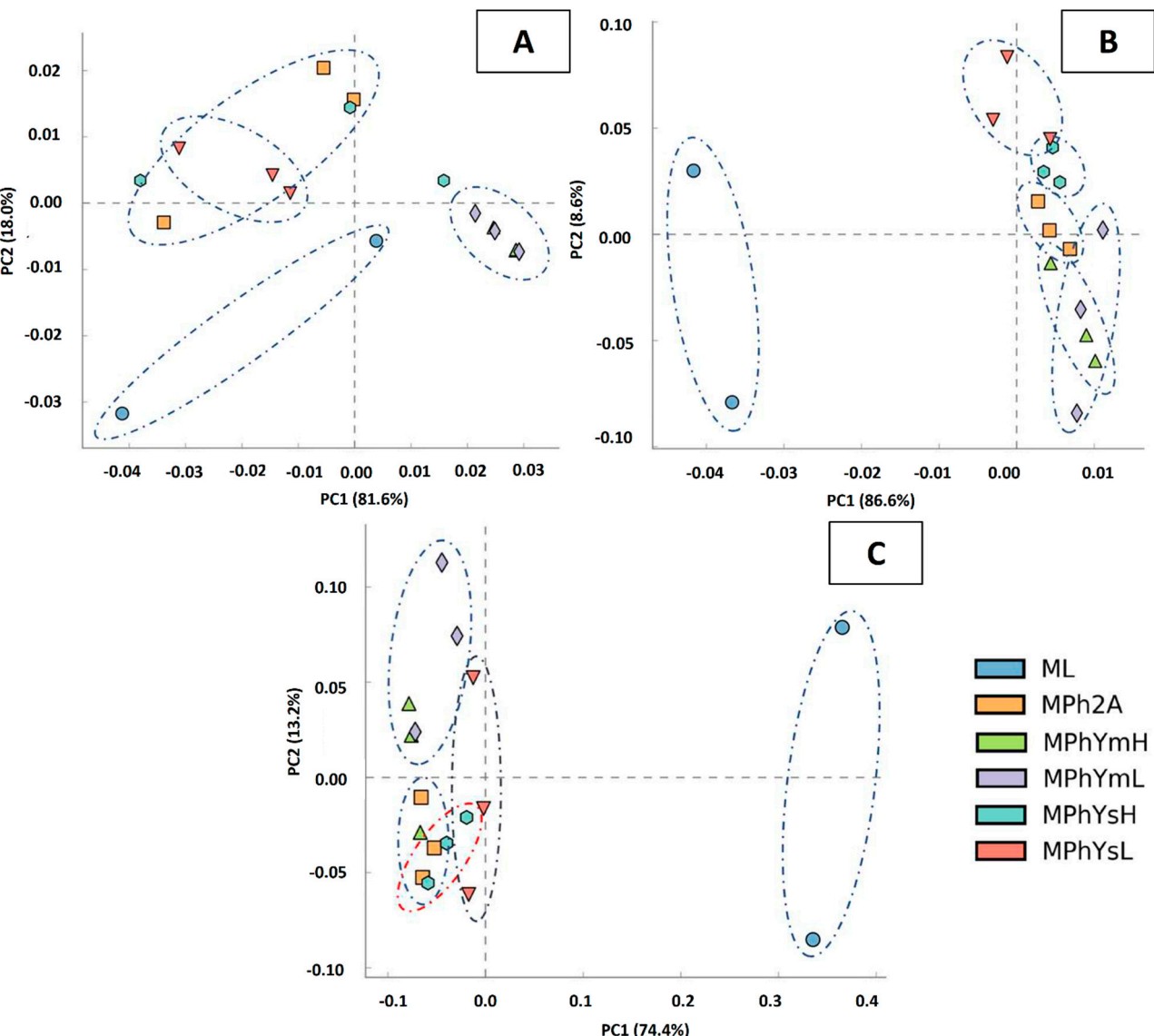

**Figure 3.** Based on MiSeq sequencing data, PCA analysis showing the distinct separation between the culturable communities developed on the three different culturing methods in relation to the culture-independent community at (**A**); phylum, (**B**); order, (**C**); genera levels. ML, culture-independent mother leaf; MPh2A, culture-dependent on R2A culture medium; MPhYmH, culture-dependent on homologous maize broth 25 mL L$^{-1}$; MPhYmL, culture-dependent on homologous maize broth 5 mL L$^{-1}$ MPhYsH culture-dependent on heterologous sunflower broth 25 mL L$^{-1}$; MPhYsL, culture-dependent on heterologous sunflower broth 5 mL L$^{-1}$.

### 3.3.3. Maize Endo-Phyllosphere Bacterial Community Composition: Culture-Independent Community

The analysis of the culture-independent bacterial community showed that Proteobacteria recorded the highest relative abundance (94.8%), followed by Bacteroidetes (3.09%), Firmicutes (1.63%), and Actinobacteria (0.1%). Among the Proteobacteria, 66.1% of the OTUs were affiliated with Gammaproteobacteria (66.1%, including orders of Pseudomonadales, 38.24%, Enterobacterales, 22.68% and Xanthomonadales, 4.09%), followed by Alphaproteobacteria (21.34%, orders of Rhizobiales 15.34% and Sphingomonadales, 6.11%) and Betaproteobacteria 7.3% (order of Burkholderiales, 7.24%). Most Bacteroidetes OTUs were affiliated with the class Flavobacteriia (3.09%), while most Firmicutes OTUs belonged to the order Bacillales (1.47%) (Tables 2 and S4).

### 3.3.4. Differential In Vitro Culturability of Bacterial Communities of Maize Endo-Phyllosphere in Response to Cross Cultivation on Homologous/Heterologous Culture Media

Differential in vitro growth at the expense of homologous versus heterologous plant media resulted in overabundance/underabundance of varying bacterial taxa on multiple levels (Figure 4). The homologous cultivation on maize plant broth selectively enriched representatives of the phylum Proteobacteria (ca. > 98.0%), in particular those belonging to Pseudomonadaceae and Moraxellaceae families. The opposite was true with heterologous cultivation on sunflower plant broth and R2A media that correspondingly enriched members of Firmicutes (ca. > 3.0%) (Figure 4). Upon in vitro cultivation on various culture media, representatives of Bacteroidetes were relatively downsized, from 2.8–3.1% to 0.01–1.66%. The taxa of Actinobacteria were relatively more abundant with heterologous cultivation (0.12–0.34%) compared to homologue cultivation (0.06–0.07%).

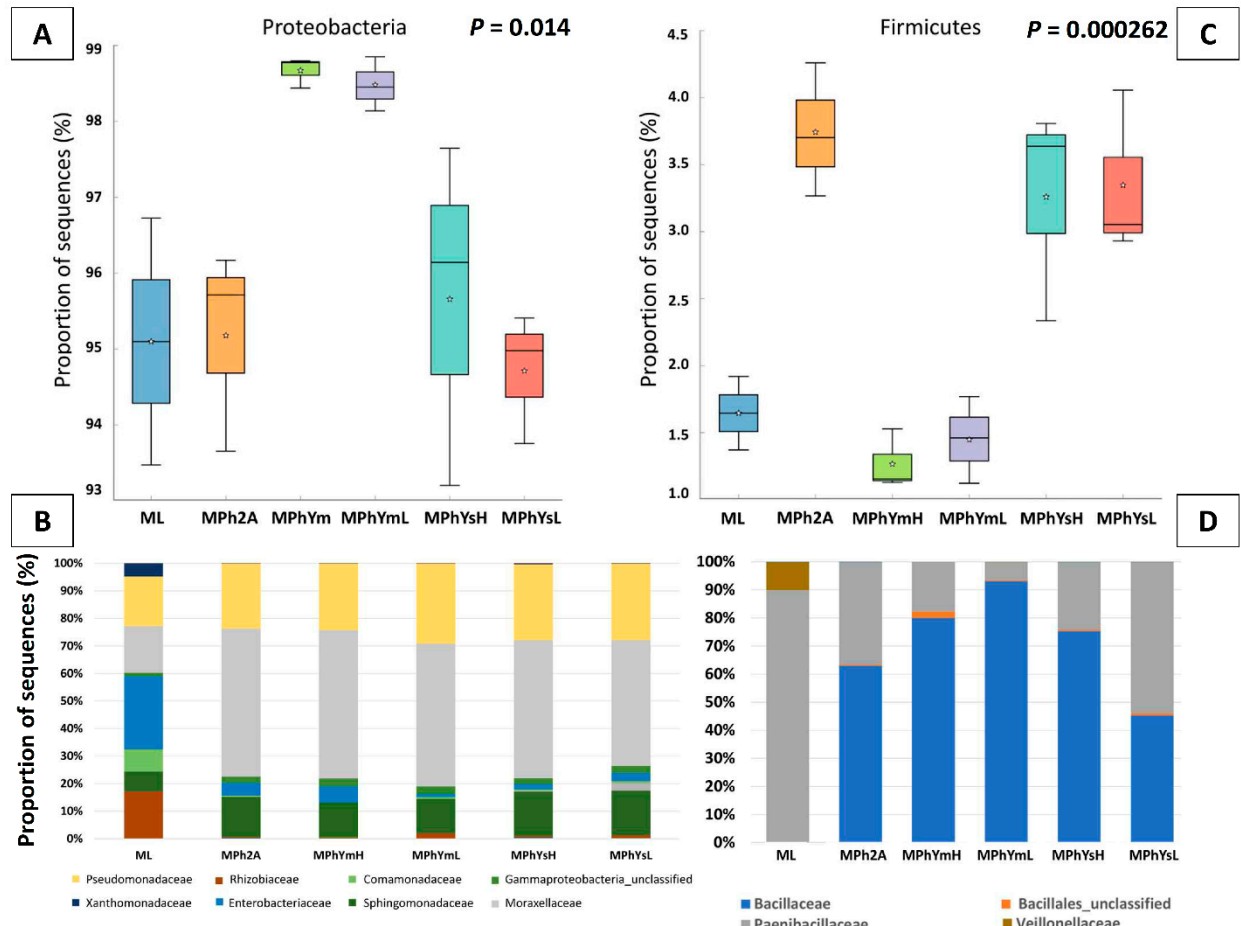

**Figure 4.** Based on MiSeq sequencing data, differential culturability of bacterial communities of maize endo-phyllosphere in response to types of culture media: Specific responses of the phylum Proteobacteria (**A**) and related families (**B**) compared to Firmicutes (**C**) and related families (**D**). ML, culture-independent mother leaf; MPh2A, culture-dependent of R2A culture medium; MPhYmH, culture-dependent on homologous maize broth 25 mL L$^{-1}$; MPhYmL, culture-dependent on homologous maize broth 5 mL L$^{-1}$; MPhYsH culture-dependent on heterologous sunflower broth 25 mL L$^{-1}$; MPhYsL, culture-dependent on heterologous sunflower broth 5 mL L$^{-1}$. *, indicates statistically significant differences.

At the family level (Figure 4; Table S5), and based on the relative abundance of the culture-independent samples, a number of families were generally downsized by in vitro cultivation, e.g., Enterobacteriaceae, Rhizobiaceae, Xanthomonadaceae, Comamonadaceae,

and Paenibacillaceae. In contrast, Moraxellaceae, and Bacillaceae were particularly enriched with cultivation on all tested culture media. While Sphingomonadaceae and Alcaligenaceae were specifically enriched on heterologous culture media, Pseudomonadaceae was enriched on homologous culture media.

## 4. Discussion

Plants are hospitable hosts that encompass their innate and interacting microbiota, forming an assemblage referred to as a "holobiont". The selective pressure exerted on such holobiont components is likely to configure the structure and function of the plant-associated microbial communities and select host-adapted microorganisms that impact plant fitness [64,65]. Such interactions between plants and their inherited microbiota are bidirectional. The host plant primarily provides packages of metabolic capabilities to recruit specific microbial taxa with desired functions at different developmental stages that lead to the adaptation of niche-specialized inhabitants [65–67]. Several reports have documented the high chemical diversity of plants that produce a discrete array of metabolites more than those produced by most other organisms [45–48,68]. To a large extent, such diverse metabolites are attributed to continual chemical modification of the basic skeletons of metabolites, and play many different roles in plant growth and development in plant response to continuing environmental changes and abiotic/biotic stresses [49].

Together, plant and microbial specialized metabolites are reciprocated and represent inherent chemical diversity of great influence in intermediating ecological interactions between organisms [50]. In other words, the chemical composition in plant organs is of a dynamic structure that is imprinted on the inherent host-microbial community in structure and function. A unique situation that compels the need to approximate and simulate comparable real-time nutritional matrices of internal plant chemical structures in any tailored culture media. This allows real-time in vitro exposure of host plant microbiota. Here, the introduction of plant-only culture media is fairly justified and qualified to simulate the plant's in situ chemical composition, both in structure and complexity [4]. As a result, this has led to deeper in vitro exposure of not-yet cultured genera, less abundant and/or hard-to-culture bacterial phyla [4,12,13,34,39,44]. To further reproduce the in vitro chemical/nutritional diversity of plants, various plant organs, e.g., leaves/roots, were incorporated into the tailored MPN-semi solid culture media [39]. PCR-DGGE analysis produced a clear distinction of bacterial taxa cultured on the homologous/heterologous leaf strips/root segments-based culture media. This underlined a divergence in the community composition of cultivable endophytes of either endo-phyllosphere or endo-rhizosphere, which signaled a certain degree of plant organ (leaves/roots) affinity/compatibility [39]. The authors attributed such preferential culturability to the nutritional makeup of the plant leaves or roots used in the preparation of plant-based culture media [40,69]. If this is the case among intra-plant organs, what will it be with inter-plant species chemodiversity? This was the rationale of the present work, mainly to experiment with and evaluate the in vitro cultivation of host plant (maize) endophytes on culture media based on their homologous (maize) or heterologous (sunflower) plant broth.

The distinct differences in the chemical analysis of maize and sunflower create a varying nutritional makeup that sculptured the culturable community structure of maize endophytes in both homologous and heterologous plant media, as evidenced by PCR-DGGE fingerprinting of the 16S rRNA gene segment. UPGMA analysis differentiated the mother culture-independent maize leaf/root apart from the culturable communities grown on the homologous (maize) and the heterologous (sunflower) plant-based medium. This conclusion was further validated by amplicon sequence data analysis of maize endo-phyllospheric bacteria. Likewise, the relative abundance of multi-million reads and derived thousands of OTUs identified a marked separation (UPGMA) of the culture-independent community followed by successive sub-clustering of the culturable bacterial communities based on homologous, heterologous, and R2A cultivation. In addition, PCA analysis asserted such separation among the culturable communities developed on the three tested

culturing methods at the phylum, order, and genus levels. Such analogy among the results of PCR-DGGE fingerprinting and those of amplicon sequence analysis is an indication of the reliability of PCR-DGGE as a routine technique for the comparison of the community composition of the environmental microbiome. Admitting that the technique has its inherent limitations, several reports have committed its reliability, being not requiring complex bioinformatics for data analysis, and presenting a general picture of the diversity of environmental microbiota when coupled with culture-dependent analysis [36,39,70,71]. In addition, and in common with NGS methods, different primer sets can be used in PCR-DGGE to address microbial communities at both the phylogenetic and functional levels [72].

Amplicon sequence data analysis of maize endo-phyllospheric bacteria revealed differential in vitro culturability of bacterial communities in response to cross cultivation on homologous/heterologous culture media. The analysis of the culture-independent bacterial community showed the highest relative abundance of Proteobacteria, with the majority of Gammaproteobacteria followed by Alphaproteobacteria and Betaproteobacteria. Secondary abundance (<5%) was in the descending order of Bacteroidetes, Firmicutes, and Actinobacteria. Among the OTUs of Gammaproteobacteria, the most common were Pseudomonadales, Enterobacterales, and Xanthomonadales. While Alphaproteobacteria were mostly represented by Rhizobiales and Sphingomonadales, Betaproteobacteria were particularly defined by Burkholderiales. Most Bacteroidetes OTUs were affiliated with the class Flavobacteria, while most Firmicutes OTUs were represented by the order Bacillales. A similar conclusion based on culture-independent analysis was reported for maize phyllospheres by Zhang et al. [8], where Gammaproteobacteria were predominant in stem microbiota (stem endosphere and xylem sap), with a gradual ascending transition in relative abundance among plant compartments: 12.5% (bulk soil), 29.2% (rhizosphere soil), 59.3% (root endosphere), and 93.2% (xylem sap). Enriched Gammaproteobacteria OTUs were dominated by Enterobacteriaceae, Erwiniaceae, Pseudomonadaceae, and Burkholderiaceae. With amplicon sequencing, Xiong et al. [65] reported that the phylloplanes of maize plants act as an important interface between the host, microbes, and the environment, and that the plant developmental stage has a much stronger influence on microbial diversity and composition. Members of Burkholderiaceae, Microbacteriaceae, Streptomycetaceae, and Rhizobiaceae were significantly enriched in phylloplanes at an early stage, and Actinobacteria, such as *Microbacterium* and *Sphingomonas*, were significantly enriched in the maize phylloplane over later plant development stages.

Similarly, the culture-dependent community was overwhelmed by the core members of Proteobacteria, accompanied by the core-satellites of Firmicutes, Bacteroidetes, and Actinobacteria. Unmistakably was the differential in vitro growth cultivation based on homologous contra heterologous plant medium that was expressed as over/less abundance of bacterial taxa on multiple levels. The homologous cultivation on maize plant broth discriminately enriched members of the phylum Proteobacteria, in particular those belonging to the Pseudomonadaceae and Moraxellaceae families of Gammaproteobacteria. On the other hand, heterologous cultivation on sunflower plant broth and R2A selectively enriched the representatives of Firmicutes. Generally, in vitro cultivation on all tested culture media relatively downsized Bacteroidetes from 2.8–3.1% to 0.01–1.66%. Actinobacteria were relatively more abundant with heterologous cultivation (0.12–0.34%) compared to homologous cultivation (0.06–0.07%). Corresponding reports on culture-dependent analysis of maize roots and shoots [73] indicated that the majority of representative endophytic bacteria belonged to Alphaproteobacteria, Gammaproteobacteria, and Actinobacteria, together with a small fraction associated with Betaproteobacteria, Bacteroidetes, and Firmicutes. Higher diversity was reported for roots, mostly belonging to Alphaproteobacteria and Betaproteobacteria, while in shoots, the majority of the endophytes belonged to Actinobacteria.

LEfSe analysis [63] was applied to determine OTUs most likely to compare differences between classes and predict OTUs groups that concisely make the differences, i.e., predict biomarkers consisting of features that characterize the existing bacterial community

structure of maize endo-phyllosphere (Figure 5A,B). The culture-independent microbiomes are strongly and widely enriched to include both clades of Alphaproteobacteria, order Rhizobiales (Family Hyphomicrobiaceae), and Gammaproteobacteria, order Xanthomonadales (Family Xanthomonadaceae). The differential growth of bacterial taxa was reported for culture-dependent analysis. With homologous cultivation on the plant broth of the host maize plant, Actinomycetales were the distinguished phylogenetic unit (Family Microbacteriaceae, Genus *Microbacterium*). A consolidating evidence on the major advantage of using "the inoculum-dependent culturing strategy, IDC," which allows bacteria to in vitro grow only at the expense of natural plant nutrients contained in the administered inoculum prepared for the homologous tested host plant organs/spheres [35]. The method comfortably and successfully recovered several isolates of endophytic Actinobacteria, representing the genera of *Curtobacterium* spp., *Plantibacter* spp., *Agreia* spp., *Herbiconiux* spp., *Rhodococcus* spp., and *Nocardioides* spp., most likely with novel species belonging to *Agreia* spp. and *Herbiconiux* spp. On the other hand, cultivation on heterologous plant broth of sunflower followed a contrasted trend and particularly enriched the family Alcaligenaceae (Genus *Achromobacter*) of Betaproteobacteria and the family Rhizobiaceae (Genus *Shinella*) of Alphaproteobacteria. Extraordinarily was the situation of cultivation on the synthetic R2A culture medium, as the most abundant and overrepresented bacterial taxa were Firmicutes, Bacilli, Bacillales, and Paenibacillaceae, represented by the genus *Brevibacillus*.

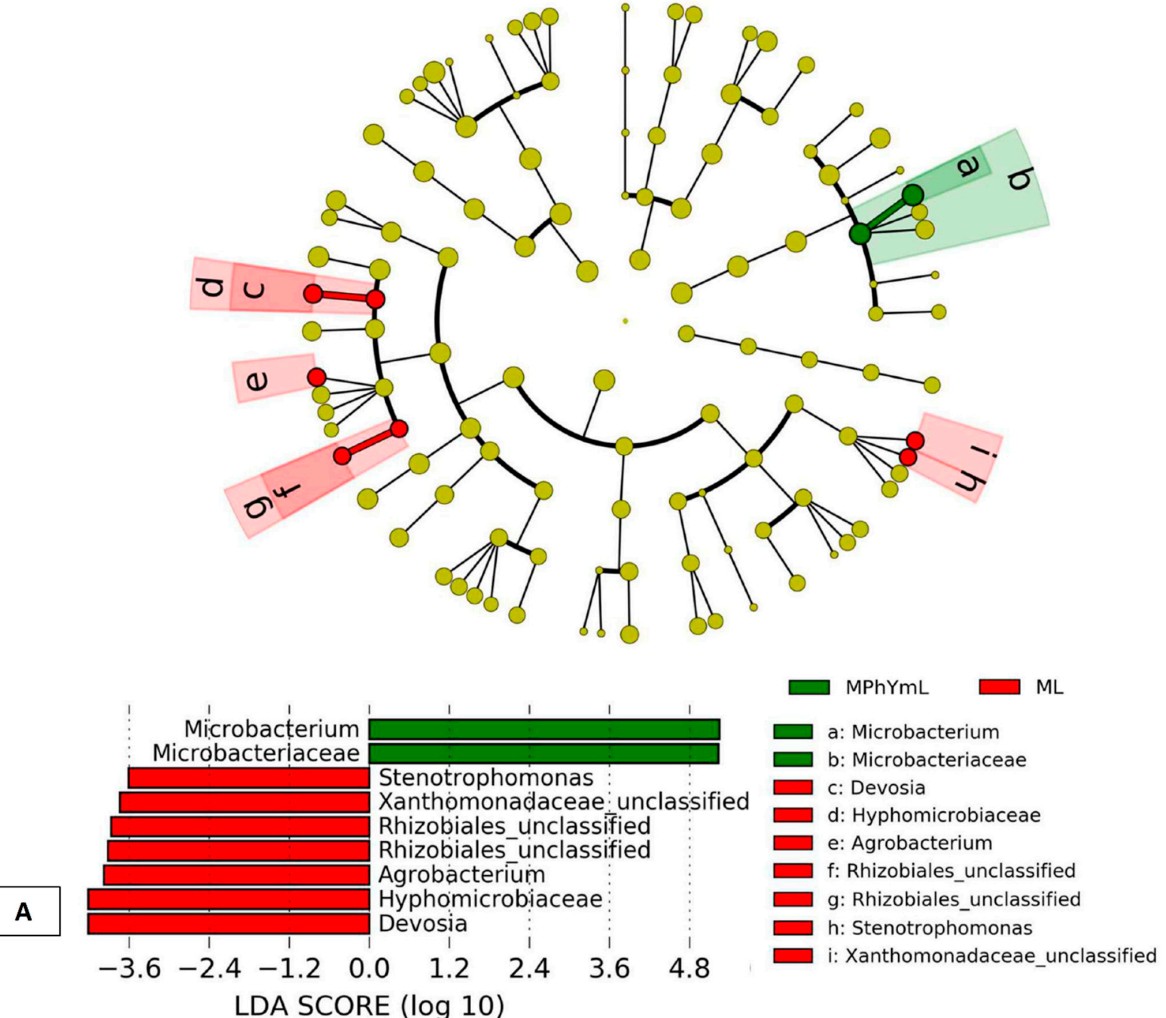

**Figure 5.** *Cont.*

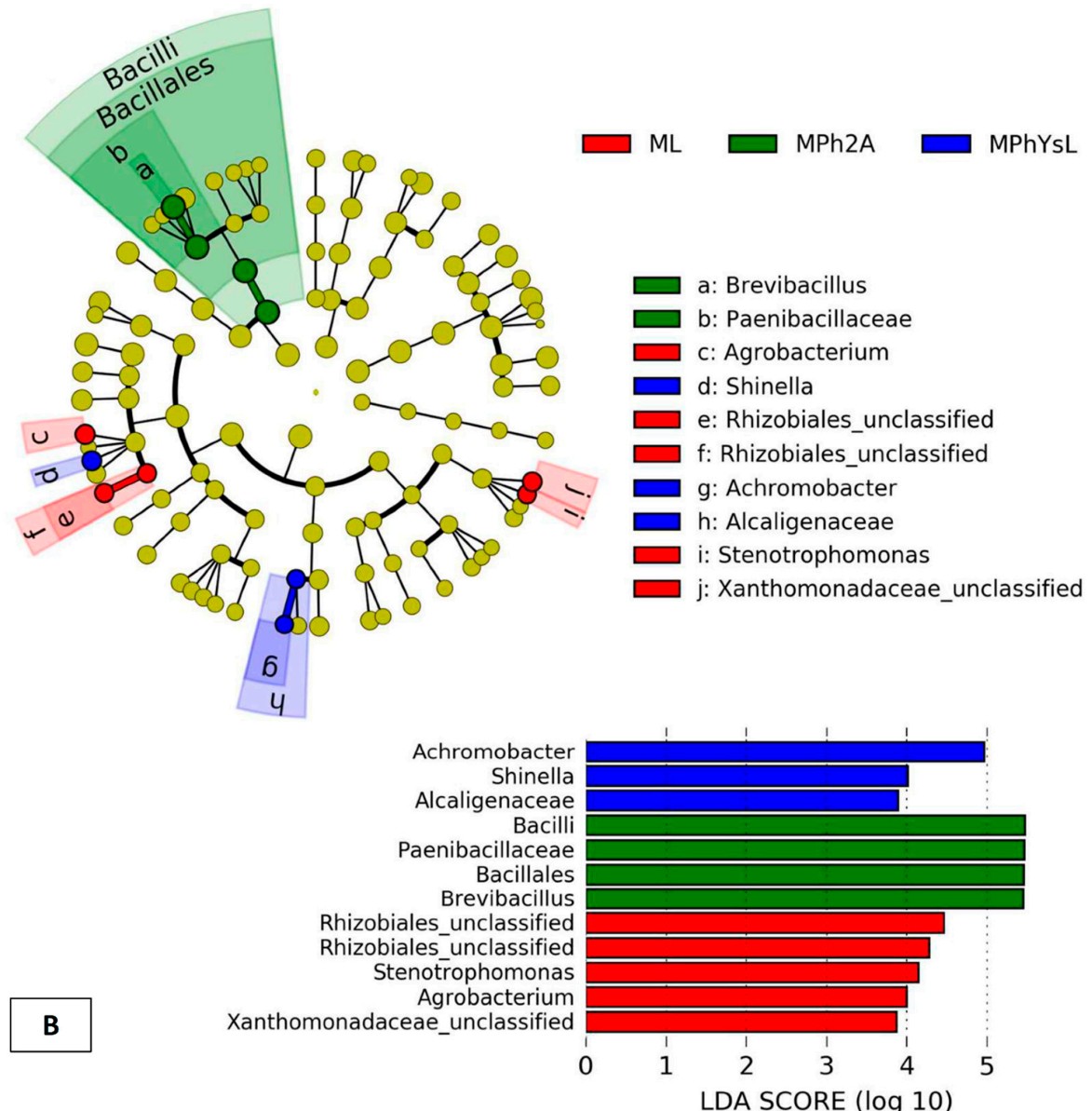

**Figure 5.** Phylogenetic dendrogram of biomarkers identified in various combinations of tested culture media: (**A**), ML and MPhYmL, MPhYmH; (**B**), ML, R2A, MPhYsL, and MPhYsH. The circles from inside to outside indicate bacterial taxonomic levels from phylum to genus. Yellow dots represent bacteria that do not vary significantly in relative abundance. Biomarker bacteria are colored according to the corresponding colors on the right. LDA scores of biomarkers, as calculated in the Kruskal–Wallis rank sum test, $p < 0.05$. ML, culture-independent mother leaf; MPhYmH, culture-dependent on homologous maize broth 25 mL L$^{-1}$; MPhYmL, culture-dependent on homologous maize broth 5 mL L$^{-1}$; MPhYsH culture-dependent on heterologous sunflower broth 25 mL L$^{-1}$; MPhYsL, culture-dependent on heterologous sunflower broth 5 mL L$^{-1}$; MPh2A, culture-dependent of R2A culture medium.

## 5. Conclusions

It is known that microbial communities establish compositional and functional alliances with their hosts and express beneficial traits capable of enhancing plant vigor. However, it remains difficult to engineer synthetic microbial communities (SynComs) that are functionally and environmentally expressed. Additionally, the thousands of pure cultures of cultivated microbiota piled in international stock centers originated from a vast range of dissimilar environments and hosts. Here, the necessity arises to explore

host/environment—compatibility and in vitro-cultivability of congruent members of plant microbiota for in situ manipulation and modification. The presented idea of an in vitro cultivation strategy based on homologous/heterologous plant-based culture media is first introduced to encourage fellow researchers to further experiment, improve, and contain limitations and delimitations of the strategy. In fact, the strategy efficiently: (a) fingerprints the complex chemical composition of host plants to facilitate real-time in vitro cultivation and lab-keeping of compatible isolates of plant microbiota; (b) opens a new horizon toward further application and widening the scope of culturomics of the plant microbiota; and (c) promotes new perspectives for preferential pairing of plant–microbe partners forward future synthetic community research and manipulating of SynComs in agriculture.

**Supplementary Materials:** The following supporting information can be downloaded at: https://www.mdpi.com/article/10.3390/d15010046/s1, Figure S1: Rarefaction curves of OTUs observed using 16S rRNA amplicon sequencing across all samples tested; Figure S2: 1, CFUs of maize endo-rhizosphere (dilution $10^{-4}$); 2, CFUs of maize endo-phyllosphere (dilution $10^{-2}$ and $10^{-3}$) developed on tested agar plates; Figure S3: Log colony forming units (CFUs) counts for cultivable endophytic bacteria recovered from the maize endo-rhizosphere (orange boxplots) and endo-phyllosphere (green boxplots) during incubation time (1, 7 and 14 days), cultivated on homologous (MPhYmH, maize broth 25 mL $L^{-1}$; MPhYmL, maize broth 5 mL $L^{-1}$) and heterologous (MPhYsH, sunflower broth 25 mL $L^{-1}$; MPhYsL, sunflower broth 5 mL $L^{-1}$) plant-based culture media as well as chemically-synthetic culture media R2A; Figure S4: UPGMA clustering of Euclidean distances of DGGE fingerprints of culture-dependent and culture-independent maize endo-rhizosphere bacterial communities. Table S1: Number of OTUs of culture-dependent endophytes of endo-phyllosphere of maize obtained for all tested culture media in three replicates, and for the original mother culture medium (culture-independent); Table S2: Nutritional profile of the dehydrated powders of tested maize (*Zea mays* L.) and sunflower plants (*Helianthus annuus* L.); Table S3: Two-way ANOVA analysis of log numbers of CFUs (data are log means $\pm$ standard error [SE], n = 5) of culturable endophytes of endo-rhizosphere and endo-phyllosphere of maize developed on the various culture media; Table S4: Relative abundance a of dominant phyla, classes and orders in all tested culture-independent and culture-dependent samples; Table S5: Relative abundance of dominant families and genera in all tested culture-independent and culture-dependent samples.

**Author Contributions:** Conceptualization, N.A.H., S.R. and M.F.; Methodology, H.E., R.A.N., E.H.N., T.R.E., M.A.H., M.E.-T. and H.H.Y.; Software, M.A.H., E.H.N. and T.R.E.; Validation, N.A.H., S.R. and M.F.; Formal Analysis, T.R.E., E.H.N., M.A.H., M.E.-T. and H.H.Y.; Investigation R.A.N., H.E., E.H.N., H.H.Y., M.A.H. and M.A.; Resources, N.A.H., S.R. and M.F.; Data Curation, H.E., R.A.N., E.H.N., T.R.E. and M.A.; Writing—Original Draft Preparation, M.F., H.E., T.R.E., E.H.N., M.A. and N.A.H.; Writing—Review & Editing, N.A.H., S.R., M.A.H., E.H.N. and T.R.E.; Visualization, N.A.H., S.R. and M.F.; Supervision, N.A.H., S.R. and M.F.; Project Administration, N.A.H., S.R. and M.F.; Funding Acquisition, N.A.H. and S.R. All authors have read and agreed to the published version of the manuscript.

**Funding:** This research received no external funding.

**Institutional Review Board Statement:** Not applicable.

**Data Availability Statement:** Sequences were submitted for deposition at the public repository Sequence Read Archive (SRA) with accession number PRJNA900715 and bioproject accession number PRJNA891051 (http://www.ncbi.nlm.nih.gov/sra (accessed on 20 September 2022)).

**Acknowledgments:** Hegazi acknowledges the generous support of the Alexander von Humboldt Stiftung for the on-going joint research projects with German partners at IGZ. We are grateful to all kinds of support provided by Eckhard George in his capacity as the research director of IGZ. Thanks are also extended to Birgit Wernitz for excellent technical support. The graphical abstract was a kind gesture of our graduates Ahmed T. Morsi and Mahmoud S. Abdelwahab.

**Conflicts of Interest:** The authors declare no conflict of interest.

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
