# Peer review of "Cross Cultivation on Homologous/Heterologous Plant-Based Culture Media Empowers Host-Specific and Real Time In Vitro Signature of Plant Microbiota"

_diversity, doi:10.3390/d15010046_

Round 1

Reviewer 1 Report

In this manuscript, the authors investigated the effect of plant-based medium in isolation of microbes from maize root and leaf tissues. Two different plant-based medium, maized- and sunflower-based extracts, and a chemical based medium were used in this progress. The authors showed that different types of medium showed a diversed enrichment for different types of bacteria at phylum, class and order level.

However, from the PCA and sequencing data, microbime isolation on sunflower-based medium and R2A medium are more close, which is slightly different with maize broth based extraction. This finding is very interesting, which suggesting utilization of homologous medium may better than chemical and heterologous-based medium. However, the culturable community are distinct with the microbiome composition in total extracts (mother leaf). Moreover, sequence proportion in each extracts could not monitor the true composition of microbiome in nature. Therefore, although the author would like to show the benefits of bacterial isolation from homologous medium, the overall data could not fully support their hypothesis.

Author Response

-We agree with the reviewer comment that the culturable communities were distinct from the natural communities detected by culture-independent analysis of mother leaf samples.

-So far, from reports in literature, it is obvious that all taxa identified by culture-independent methods are not successfully recovered/cultivated  by present culture-dependent methods. And, still extensive efforts are to be exercised to improve in vitro culturing methods in order to truly simulate the in situ microbiome composition.

And, the present manuscript is considered among such efforts to improve in vitro culturability and expose diversity of  in situ-plant microbiome (i.e. this is the main hypothesis of the presented manuscript)

- We acknowledge that the presented manuscript is introducing the idea of homologous cultivation for the first time, and that  requires further experimentation and improvement by fellow-researchers.

However, we draw the kind attention of the reviewer that the manuscript is representing original data that deserve publication, regarding the evident differences in vitro growth on homologous versus heterologous cultivation that justify the consideration of the method for further improvement and application; for example:

·         Homologous cultivation selectively enriched the phylum Proteobacteria , in particular Pseudomonadaceae and Moraxellaceae families; contrary to heterologous cultivation and R2A media that enriched members of Firmicutes (see Figure 4).

·         LEfSe analysis showed the differential growth of bacterial taxa on various culture media. With homologous cultivation, Actinomycetales were the distinguished biomarkers (Family Microbacteriaceae), compared to  heterologous cultivation that enriched the family Alcaligenaceae of Betaproteobacteria and the family Rhizobiaceae of Alphaproteobacteria. On the other hand, R2A overrepresented bacterial taxa of Firmicutes, Bacilli, Bacillales, and Paenibacillaceae (see Figure 5a,b).

Reviewer 2 Report

Dear authors, I have read the manuscript "Cross cultivation on homologous/heterologous plant-based culture media empowers host-specific and real time in vitro signature of plant microbiota".

There are some suggestions that I consider will improve your manuscript.

As a general comment I suggest you to rewrite the sentences that contain personal addressing (e.g. we) in an impersonal form. Avoid long sentences, more than 4-5 rows and better split them in two sentences with one or two ideas explained in each.

The introduction is very long, but explains extensively the background of your work. I suggest you to move some parts, if you consider, to the Discussion section. Overall, this section is well written.

Mat and Meth section - I don not understand the necessity for the (Graphical Abstract) in 2.1 Title. The idea of a graphical abstract is very good if you can provide one.

Please add a sentence or two to explain which parts of the protocol presented in this section is created by the authors and which ones are from the literature (sub-sections 2.1-2.4). This will help other researchers to replicate your study or to create new research/laboratory techniques.

Results section - I have not found supplementary Tables and Figures. You reference large parts of your work to these tables. Please provide them.

Pay attention to text formatting (e.g. line 336) and line numbers present in Table 1. Expand your interpretation of both Tables and Figures according to their number. Subsection 3.1. refers to two table and two figures. You need to present the results in an expanded form.

Same observation of sub-sub-section 3.3.1. Total OTUs obtained. Present some of the affiliations.

What is the meaning of a and b from Table 2. Caption?

The idea of marking with colors the increases and decreases is interesting and good. What yellow means?

3.3.4. - I don not think that sentence from lines 428-431 is necessary.

Discussion section - this section present a mix of results and discussions. Please move the results to their section and leave only the text that compare your results with other studies. Do not make in this section references to previous presented tables/figures.

Figure 5 and its explanation should be moved to results section. There will have a greater impact.

Conclusion section - In this section you need to present your main findings, to point them in separate sentences. In this form they are too general.

Overall, I like the idea of your manuscript and you present numerous information. I suggest you to make a table to present a condensed form of your Mat and Meth section.

Author Response

-Thanks. We tried to make it short as much as possible

-The introduction section is somewhat long as it broadly treats the current status of in vitro culturability of plant microbiota from various aspects: the microbiota, the host plants, and the pertinent methods of in vitro cultivation.

-Additionally, the main points was spotlighted as well in the discussion section

Agree, adjusted within the text (M & M) and  highlighted in yellow

The supplementary material is uploaded again

-Thanks, format adjusted

-The text is adjusted to interpret the numbers in the related figs and tables; we clearly indicated  the main finding related to the effects of incubation time, culture media and interactions

The text is adjusted

Apologize for the typo while preparing the manuscript, both a and b letters were deleted

Apologize, as the yellow marking was highlighted during preparation of the manuscript, and mistakenly was left; and now removed

Agree, the sentence is deleted

Agree.

-During writing the manuscript, we have tried to avoid that.

- we have tried to follow the advice; the only and major exception was the results (and Fig 5a,b) of  LEfSe analysis. As it was of importance to expose these results in correlation with those of MiSeq analysis in the discussion. And, to avoid repetition in results and discussion

As mentioned above, we request the understanding of the reviewer to keep it in discussion.

Thanks.

If we understood correctly the point of view of the reviewer, we purposely designed the  graphical abstract to  illustrate the work flow and in general the methods involved.

Thanks for raising the idea, as this encourages us  to proceed with our plan  to draft a manuscript on our multiple protocols of plant-only-based culturing strategies developed along the last decade, to be submitted to “Methods and  protocols, MDPI”

Reviewer 3 Report

In my opinion, the topic of the manuscript entitled "Cross cultivation on homologous/heterologous plant-based culture media empowers host-specific and real time in vitro signature of plant microbiota" is practical and interesting and will be interesting for the potential reader.

The formal style should be used, so forms like “we” ought to be avoided. English also be needs to carefully checked and corrected, preferably by a native speaker.

Introduction should be shortened. 

Line 16, 34, 425 - The phrase “in vitro” should be italicized. Please, totally check that in the text.

Figure 1, 3 and 4 should have better quality

In the Conclusion part, the authors do not provide details of any limitations of this study and recommendations for future perspectives.

Author Response

Thanks and appreciate

-This point was raised as well by reviewer #2; and relevant cases were adjusted all over the text.

-We hope that the revised  manuscript is  substantially improved through extensive edits; and that the  English language was improved as well based on the valuable comments of the group of reviewers, together with the  extensive revision by senior authors  of long experience with publishing in English language

-          We tried our best; however, we request the understanding of the reviewer, as indicated by reviewer #2, as the introduction treats multiple topics of in vitro cultivation of the plant microbiome including the many limiting factors, pertinent methods, and  in particular host plant metabolism and interaction.

Agree, adjusted all over the text

Agree.

We hope that the attached individual figs, not those inserted within the main text, are of good enough. Besides, Fig 4 was further improved

The conclusion part  is adjusted.

Round 2

Reviewer 1 Report

Dear Editor,

Although the authors performed experiments to show the idea that using plant tissue based medium improves the isolation of plant associated microbes, the results could not support their claim since the microbiome composition of all isolation methods are far away from the microbe in nature. It is difficult to judge whether this method could help to understand the interaction between host and microbes comparing with other isolation. The plant-tissue based isolation may also have their own biases.

Author Response

As mentioned in our response to the comments of the reviewer during the first revision, we stress that none of the present in vitro culturing methods fully recover all of the taxa resolved by in situ by culture independent methods. And we completely agree that the presented methods, like other methods, has its own inherent errors and limitations. Therefore, we stressed all over the manuscript that the introduced method of in vitro cultivation on homologous plant-based culture media offers an additional new avenue for exploring the plant microbiota. 

Reviewer 2 Report

Dear authors, 

The new form of your manuscript present multiple improvements. It present your reseach better.

Author Response

thanks and appreciate